# Ubiquitylation of ABA Receptors and Protein Phosphatase 2C Coreceptors to Modulate ABA Signaling and Stress Response

**DOI:** 10.3390/ijms22137103

**Published:** 2021-07-01

**Authors:** Alberto Coego, Jose Julian, Jorge Lozano-Juste, Gaston A. Pizzio, Abdulwahed F. Alrefaei, Pedro L. Rodriguez

**Affiliations:** 1Instituto de Biología Molecular y Celular de Plantas, Consejo Superior de Investigaciones Científicas-Universidad Politécnica de Valencia, ES-46022 Valencia, Spain; cgalberto63@hotmail.com (A.C.); jjvrf2@gmail.com (J.J.); lojujo@ibmcp.upv.es (J.L.-J.); gapizzio@gmail.com (G.A.P.); 2Zoology Department, College of Science, King Saud University, Riyadh 11451, Saudi Arabia; afrefaei@ksu.edu.sa

**Keywords:** abscisic acid, ABA receptor, clade A PP2C, E3 ubiquitin ligases, PYR/PYL/RCAR, RBR E3 ligase, CRL3, CRL4, RING E3 ligase, PUB E3 ligase

## Abstract

Post-translational modifications play a fundamental role in regulating protein function and stability. In particular, protein ubiquitylation is a multifaceted modification involved in numerous aspects of plant biology. Landmark studies connected the ATP-dependent ubiquitylation of substrates to their degradation by the 26S proteasome; however, nonproteolytic functions of the ubiquitin (Ub) code are also crucial to regulate protein interactions, activity, and localization. Regarding proteolytic functions of Ub, Lys-48-linked branched chains are the most common chain type for proteasomal degradation, whereas promotion of endocytosis and vacuolar degradation is triggered through monoubiquitylation or Lys63-linked chains introduced in integral or peripheral plasma membrane proteins. Hormone signaling relies on regulated protein turnover, and specifically the half-life of ABA signaling components is regulated both through the ubiquitin-26S proteasome system and the endocytic/vacuolar degradation pathway. E3 Ub ligases have been reported that target different ABA signaling core components, i.e., ABA receptors, PP2Cs, SnRK2s, and ABFs/ABI5 transcription factors. In this review, we focused specifically on the ubiquitylation of ABA receptors and PP2C coreceptors, as well as other post-translational modifications of ABA receptors (nitration and phosphorylation) that result in their ubiquitination and degradation.

## 1. Introduction

The Ub-26S proteasome (UPS) and vacuolar degradation pathways play fundamental roles in the regulation of protein half-life and control of protein homeostasis in plant cells [1,2,3]. In particular, plant response to abiotic stress is modulated by a myriad of UPS components as well as non-26S proteasome endomembrane trafficking pathways [4,5,6,7]. The UPS first catalyzes the attachment of Ub molecules to selected targets, and this promotes their degradation by the multiprotease 26S proteasome complex [2]. The E1 (Ub-activating enzyme, UBA), E2 (Ub-conjugating enzyme, UBC), and E3 (Ub ligase) cascade is involved in conjugating Ub to substrate proteins [2,7]. The Arabidopsis genome contains 2 UBAs, 37 UBCs, and more than 1400 E3 ligases, which are divided in monomeric (HECT, RING, RBR, U-box type E3s) and multimeric E3 ligases [7]. These latter E3s are assembled using Cul1, Cul3a/3b, or Cul4 scaffold proteins, which bind the RING box-1 (RBX1) protein for recruitment of the E2 UBC in the cullin (CUL)-RING E3 ubiquitin ligase (CRL) complex. Additionally, multimeric CRLs require substrate adaptors for recognition of the target proteins. Cul1-based E3 complexes are usually referred to as SCF (for SKP1-Cullin-F box), whereas Cul3- and Cul4-based complexes are known as CRL3 and CRL4, respectively. On the other hand, the endosomal trafficking pathway plays a key role in the turnover of membrane proteins and requires components of the endosomal sorting complex required for transport (ESCRT) machinery, such as the FYVE domain-containing protein 1 (FYVE1)/FYVE domain protein required for endosomal sorting 1 (FREE1) and vacuolar protein sorting 23A (VPS23A) members of the ESCRT-I complex, and ALG-2 interacting protein-X (ALIX) associated protein of ESCRT-III. In this review, we focused on protein homeostasis of ABA receptors and PP2C coreceptors, as well as E3 ligases and ESCRT components involved in the regulation of their turnover (Table 1 and Table 2).

## 2. Ubiquitylation of ABA Receptors

Ubiquitylation of ABA receptors was reported by Bueso et al. and Irigoyen et al. [8,9] using affinity purification of ubiquitinated proteins with p62-agarose and immuno-analysis of protein extracts prepared from MG132-treated plants, which expressed HA-tagged versions of PYR1, PYL4, and PYL8. Additionally, these authors identified E3 ubiquitin ligases that target ABA receptors in the plasma membrane (PM), namely RSL1 for PYR1 and PYL4 [8], or in the nucleus, through the multimeric CRL4 E3 ligase that uses DET1 and DDB1-ASSOCIATED1 (DDA1) as part of a substrate adaptor module for recognition of PYL8 and PYL9 [9]. The RSL1- and CRL4^DDA1^-dependent ubiquitination pathways point to two different mechanisms for receptor degradation: endosome-mediated vacuolar degradation and 26S proteasome, respectively. What are the main features of these E3 ligases that define two pathways for receptor degradation and why should ABA signaling be doubly regulated? The answer to this question requires an overview on the properties of the above E3 ligases, ABA receptors, and ABA signaling.

RSL1 is anchored tomembrane through a C-terminal transmembrane domain (TMD) and originally was annotated as a RING-type E3 ligase [8]; however, further inspection of the RSL1 protein reveals three putative RING domains in tandem, named as RING1-IN BETWEEN RING (IBR)-RING2, and accordingly, RSL1 belongs to the RBR-type E3 ligase family [10,11,12]. Ubiquitylation in the PM triggers endocytosis of integral membrane proteins (nutrient transporters, ion channels, signaling receptors) or proteins associated to the PM [13,14,15]. The turnover of ABA receptors in the proximity of the PM likely plays an important role in regulating ABA effects on different ion and water transporters [16,17,18]. Upon endocytosis, ABA receptors follow sorting through the ESCRT machinery and finally vacuolar degradation [8,19,20,21]. At least three components of the ESCRT machinery, namely VPS23, FYVE1/FREE, and ALIX (Figure 1), have been involved in the transit of ubiquitinated ABA receptors through the ESCRT machinery. Knock-down mutations in these ESCRT components lead to attenuated degradation of ABA receptors and enhanced ABA signaling [19,20,21]. Given that ABA receptors belong to a gene family, it is not surprising that RSL1 also belongs to a gene family, in this case composed of at least 9 additional members, which were named RFA1 to RFA9 for RING finger ABA-related 1-9 [8]. RFA6 to RFA9 contain a TMD at the C-terminus of the proteins, which suggests they are membrane-localized as RSL1, although their expression levels are much lower than RSL1 [11]. In contrast, RFA1 to RFA5 lack a TMD and show distinct subcellular localizations. For example, Fernandez et al. [11] found that RFA1 is localized both in the nucleus and cytosol, whereas RFA4 shows specific nuclear localization and promotes nuclear degradation of ABA receptors. Therefore, members of the RSL1/RFA family interact with ABA receptors in the PM, cytosol, and nucleus, targeting ABA receptors for degradation via the endosomal/vacuolar RSL1-dependent pathway or 26S proteasome. Thus, the complexity of the ABA receptor family is mirrored in the partner RBR-type E3 ligases.

In addition to the PM signaling events, ABA has a dramatic transcriptional effect (affecting approximately 10% of the plant transcriptome) mediated by ABA receptors, such as PYL8, which can be localized in the nucleus; therefore, nuclear degradation of ABA receptors mediated by nuclear E2-E3 ligases is a necessary feedback mechanism to modulate the transcriptional response to ABA. A pioneering finding regarding this subject was the discovery that CRL4 E3 ligase regulates PYL8 degradation using DDA1 as part of the substrate adapter module, which carries out the recognition of the target protein [9]. Interestingly, the CRL4 substrate receptors DW1 and DW2 were previously reported to mediate ABI5 degradation by CRL4 complexes [22], as well as the DWD protein named ABD1 [23]. CRL4 ligases are complex multimeric enzymes, which are based on the CUL4 scaffold, and for example, in Arabidopsis the associated COP10-DET1-DDB1 (CDD)-related complex has been also identified in these E3s [9]. Moreover, the DDA1 protein, as part of the CDD complex, provides recognition of the substrate to be ubiquitinated. Using DDA1 as bait in the yeast two-hybrid (Y2H) screen, Irigoyen et al. [9] found interaction with PYL4, PYL8, and PYL9. Using HA-tagged PYL8 lines, it was demonstrated that DDA1 overexpression promotes PYL8 protein degradation through the 26S proteasome and reduces plant sensitivity to ABA. Interestingly, in the presence of ABA, the CRL4^DDA1^ complex cannot promote the degradation of PYL8, which suggests that ABA protects PYL8 from CRL4^DDA1^-mediated degradation either by affecting the CDD-DDA1 complex or by forming PYL8 complexes with PP2Cs that are resistant to degradation. Recent structural insights into DDA1 function as a core component of the CRL4-DDB1 complex suggest that DDA1 might use its flexible C-terminal region to recruit substrate receptors known as DCAF (DDB1 and Cullin4-associated factors) to facilitate substrate recruitment or modulate the topology of the substrate-loaded CRL4 E3 complex [24].

PYL8 is a singular ABA receptor that plays a non-redundant role to regulate root ABA perception through a non-cell-autonomous mechanism; compared to other ABA receptors, PYL8 is unique in showing ABA-induced stabilization and predominant nuclear localization [27,28]. ABA signaling regulates root growth and stress recovery, and is critically required for root hydrotropism, an adaptive response designed to facilitate soil exploration in the search for water. Moreover, different environmental cues, such as salinity and nutrient availability, induce root adaptations that are mediated by both ABA-dependent transcriptional responses and effects on PM ion transporters. Regulation of root growth by ABA is closely connected with hydrotropism, as the hydrotropic response involves asymmetric ABA signaling in the root cortex through the PYR/PYL/RCAR-PP2C-SnRK2 core signaling pathway [29,30]. It has been recently reported that low ABA increases (perceived in roots by certain monomeric ABA receptors asPYL8) can inhibit clade A PP2Cs such as ABI1, which facilitates activation of SnRK2.2 and relieves inhibition of PM ATPase AHA2 [30]. This ABA increase might occur asymmetrically in the two faces of the root, giving rise to asymmetric H^+^ extrusion at an early stage of the hydrotropic response. This determines differential root growth and likely involves selective accumulation of ABA in the convex (facing dry soil) side of the root. Given that ubiquitylation of PYL8 is reduced by ABA [9,28], the regulation of the turnover of important receptors for ABA signaling in root has a deep impact on plant physiology and water stress adaptation.

From the publications of Irigoyen et al. and Bueso et al. [8,9], a number of additional E3 ligases that target ABA receptors have been reported. Interestingly, the next discovery was achieved in rice, where Lin et al. [31] identified the APC/C (anaphase promoting complex/cyclosome) as a regulatory module of ABA and GA signaling pathways. The APC/C complex contains a distant cullin member called APC2 and a RING RBX1-like protein called APC11, and additionally the Tiller Enhancer (TE) protein that activates the E3 complex through the recognition of different targets. Although most of the substrates targeted by APC/C are related to the control of cell cycle progression, other substrates are emerging. Thus, TE, which represents the Cdh1 ortholog in rice, acts as a substrate recognition factor and targets some rice ABA receptors (OsPYLs) for degradation [31]. Specifically, TE recognizes the destruction box (D-box: RxxLxxxxN/D/E) that is present in OsPYLs (located in the β4 sheet) and promotes their degradation by the proteasome. In particular, recombinant OsRCAR2, OsRCAR9, and OsRCAR10 proteins were very stable in cell-free degradation assays of *te* mutant plant extracts [31]. The authors focused on OsRCAR10, which shares the highest homology to AtPYL1 and AtPYR1, and generated an anti-R10 antibody. Thus, Lin et al. [31] demonstrated that OsRCAR10 over-accumulates in *te* mutant plants, whereas the receptor protein level was reduced in TE overexpressing lines. However, cell free degradation assays in the *te* mutant also suggest that besides the APC/C^TE^ complex, other E3 ligases are involved in the degradation of rice ABA receptors [31].

The development of antibodies against PYL proteins enables the monitoring of the effect of loss-of-function and gain-of-function mutants of E3 ligases on endogenous levels of their targets. For example, other monomeric or putative subunits of multimeric E3 ligases that interact with ABA receptors have been described in Arabidopsis [32,33,34]. However, a detailed evaluation of each E3 contribution to the regulation of endogenous receptor levels in different tissues and developmental stages has not been reported yet. For example, the F-box RIFP1 (forming part of the SCF^RIFP1^ complex) interacts with RCAR3/PYL8, and in cell-free degradation assays, His-tagged recombinant RCAR3/PYL8 was more stable in RIFP1 background than in wild type (WT); however, endogenous RCAR3/PYL8 levels were not analyzed in RIFP1 [33]. The DWD protein RAE1 (putative substrate receptor of CUL4-DDB1 E3 ligase) interacts with RCAR1/PYL9, and in cell-free degradation assays, overexpression of RAE1 leads to enhanced degradation of His-tagged RCAR1/PYL9 compared to WT [32]. Altogether, these works illustrate the importance of substrate adaptors of either SCF or CRL4 complexes to regulate the homeostasis of ABA signaling components. Finally, the PUB22/23 U-box E3 ligases interact with PYL9 and the in vivo decay of myc-tagged PYL9 was reduced in the PUB22 PUB23 double mutant compared to WT [34].

Recently, the availability of antibodies against PYR1 and PYL4 has enabled the analysis of endogenous receptor levels in loss-of-function mutants impaired in the E3 ligases RFA1 and RFA4 [11]. Thus, both PYR1 and PYL4 ABA receptors were higher in the RFA1 RFA4 double mutant than in wild-type plants. Moreover, UBC26 was identified as the cognate nuclear E2 enzyme that interacts with the RFA4 E3 ligase, and loss-of-function UBC26 alleles also showed higher levels of PYL4 compared to the WT [11]. These results have revealed the presence of UBC26-RFA4-PYL complexes that mediate the degradation of ABA receptors in the nucleus. This pathway converges, for example, with the function of the CRL4^DDA1^ complex, which also mediates the nuclear degradation of ABA receptors, although the CRL4^DDA1^-mediated degradation of PYL8 seems to work only at reduced ABA levels. The study of human E3 ligases has coined the concept of “E3–E3 team tagging”, which implies that different types of E3s can act successively on a common target [35]. For example, a reciprocal role for RBRs–CRLs has been described for cooperative ubiquitylation of the same client substrates [35]. Therefore, chances exist for further studies on the cooperative role played by the different E3 ligases that target ABA receptors.

## 3. Regulation of Ubiquitylation by Post-Translational Modifications

Mass spectrometry analysis of ABA receptors and ABA-activated SnRK2s has identified different post-translational modifications [36,37,38]. Such modifications seem to regulate protein activity in a dual manner, namely by affecting biochemical function of the signaling proteins and also by promoting ubiquitylation and protein degradation. For example, ABA induces phosphorylation and activation of SnRK2s by B2/B3-type RAF kinases, but as a side effect, degradation of SnRK2s in promoted by ABA [37]. Interestingly, when phosphorylation of SnRK2s is abolished in the high-order RAF mutants (namely *OK100-oct* and *OK100-nonu* mutants), ABA-induced SnRK2 degradation is prevented [37]. For ABA receptors, phosphorylation usually downregulates their activity and additionally promotes their degradation [38,39,40]. For example, phosphorylation of ABA receptors by Arabidopsis EL1-like (AEL) casein kinases leads to their destabilization and the suppression of the ABA response [39]. CEPR2 also phosphorylates and accelerates the degradation of ABA receptors [40]; thus, the receptor-like kinase CEPR2, rather than participating in a phosphorylation cascade for signaling, exerts direct control of receptor activity and stability. In ABA receptors, Castillo et al. [36] identified tyrosine nitration and S-nitrosylation at cysteine residues. Tyrosine nitration reduced receptor activity, whereas S-nitrosylated receptors were active and fully inhibited PP2C activity [36]. The inactivation of PYR/PYL/RCAR receptors by tyrosine nitration is likely linked to their degradation, as those receptors with nitrated tyrosine were also polyubiquitinated [36]. Tyrosine nitration is triggered by the short-lived oxidant molecule peroxynitrite, combination of NO with superoxide radical, which suggests a rapid mechanism to inhibit ABA signaling by nitric oxide, in agreement with the ABA-hypersensitive phenotype of NO-deficient plants [41].

## 4. Ubiquitylation of Clade A PP2Cs in the Context of ABA Signaling

ABA receptors play a double inhibitory role on clade A PP2Cs because, in addition to inhibiting PP2C activity, the degradation of certain PP2Cs promoted by E3 Ub ligases is also facilitated by the formation of ternary complexes with ABA and ABA receptors [25,42]. The regulation of protein activity and stability of clade A PP2Cs is crucial in ABA signaling because these proteins are key repressors of the pathway [43]. Thus, the ubiquitylation of clade A PP2Cs enhanced by ABA, and ABA receptors help in the activation of the ABA response. According to recent discoveries in the ABA pathway, we can distinguish several steps in ABA signaling: (i) PP2C inhibition by ABA receptors in an ABA-dependent manner, (ii) activation of subfamily III SnRK2.2/3/6 kinases (SnRK2s) by phosphorylation of B2 and B3 RAFs at Ser171 and Ser175 residues (SnRK2.6/OST1 nomenclature), (iii) amplification of SnRK2 activation by intermolecular trans-phosphorylation of SnRK2s by already active SnRK2s, (iv) ABA response in the PM and nucleus through SnRK2-dependent phosphorylation of different targets and lack of PP2C inhibition on them, including ion transporters and ABRE-binding factors (ABFs), (v) ABF-mediated induction of clade A PP2Cs as a negative feedback mechanism that involves the accumulation of newly synthesized PP2Cs, and (vi) finally the resetting to basal PP2C protein levels (PP2C proteostasis). Different E3 ubiquitin ligases play a major role for the dynamic regulation of protein levels of the above-described components, and concerning PP2Cs, this contributes to both the activation and resetting of ABA signaling. For other components such as ABA receptors and ABA-activated SnRK2s, protein degradation contributes to desensitization of ABA signaling [4,44,45]. The transcriptional response to ABA is strongly dependent on ABFs, which also mediate the rapid induction of PP2Cs to modulate ABA signaling. Finally, when stress decays, E3 ligases are critical to return clade A PP2Cs to their basal protein levels under non-stress conditions.

## 5. ABA- and/or Receptor-Dependent Degradation of PP2Cs

The pioneering work of Kong et al. [42] demonstrated that clade A PP2C ABI1 is degraded by the 26S proteasome. To determine which E3 ligases target ABI1 for degradation, the authors tested as putative preys in Y2H assays several candidates reported to interact with ABI5 (such as DWA1/2 or KEG), others involved in stress response (such as RGLG1/2 and SDIR1), and 23 plant U-box (PUB) E3 ligases. The authors found that PUB12 and PUB13 interact with ABI1 using different assays, and peptides of PUB12 and PUB13 could be recovered in coIP/MS proteomic analyses of immunoprecipitated ABI1. PUB12 and 13 promote the ubiquitylation of ABI1 in a receptor-dependent manner, which is strictly dependent on exogenous ABA for the dimeric PYR1 receptor or enhanced for the monomeric PYL4/PYL9 receptors [42]. Using specific antibodies against ABI1, the authors found that the ABI1 protein level was higher in the PUB12 PUB13 double mutant than in WT, and that ABA enhanced the degradation of ABI1. PUB12 and 13 were previously reported to ubiquitinate the receptor-like kinase FLS2 located in the PM, which mediates flagellin-induced FLS2 endocytosis/degradation [46]. Flagellin induces recruitment of PUB12 and PUB13 to the FLS2 receptor complex through BAK1-mediated phosphorylation of the E3 ligases. By analogy, it is tempting to speculate that ABI1 might be ubiquitinated by PUB12/13 in the proximity of the PM when receptor-ABA-ABI1 complexes are formed.

Since PUB12/13 do not interact with other clade A PP2Cs such as ABI2, HAB1, or PP2CA, the identification of additional E3 ligases that target them was expected. Using the RGLG5 RING-type E3 ligase as bait in Y2H assays, Wu et al. [47] found interaction with PP2CA, and additionally with ABI2 and HAB2. From the five-member RGLG E3 ligase gene family, also RGLG1 (but not RGLG2) showed interaction with PP2Cs. Specific antibodies against PP2CA were raised using the N-terminal part of the protein as antigen. Thus, it was demonstrated that endogenous PP2CA follows 26S proteasome degradation, which is enhanced by ABA, and the turnover of PP2CA was markedly diminished in RGLG1 *amiR*-RGLG5 double mutant compared with the WT [47]. Therefore, both RGLG1/5 E3 ligases target PP2CA for degradation in an ABA-enhanced manner, but the mechanism was unknown at the time because both RGLG1 and RGLG5 promoted ubiquitylation of PP2CA in vitro in the absence of ABA or PYL4 [47]. In vivo analyses and subcellular localization studies were required to unveil how ABA enhances the ubiquitylation of PP2CA by RGLG1 [25]. First of all, myristoylated RGLG1 is localized in the PM, whereas PP2CA is predominantly localized in the nucleus; however, ABA inhibits the myristoylation of RGLG1 and promotes cycloheximide-insensitive translocation to the nucleus, where RGLG1 recognizes receptor-ABA-phosphatase ternary complexes [25]. As occurs with PUB12/13, the RGLG1 E3 ligase can interact with both PP2CA and certain monomeric receptors, which can facilitate the recognition/ubiquitylation of the phosphatase by RGLG1. The connection of PUB12/13 and RGLG1 E3 ligases with ternary phosphatase complexes suggests that at least a pool of them can be recognized for PP2C ubiquitylation and degradation [48].

Different E3 ligases can target the same substrate depending on the physiological and cellular context, and the integration of distinctive types of E3 ligases has been reported to act simultaneously on the same client substrates [49,50]. The affinity of different E3 ligases for their substrates, although rarely measured, can differ, and this will determine preferential E3-substrate interactions according to substrate protein levels, which notably oscillate for clade A PP2Cs in response to ABA. Concerning PP2CA, several E3 ligases that mediate its ubiquitylation have been reported [26,47,51]. Using specific antibodies against PP2CA, both Wu et al. [47] and Julian et al. [26] showed that endogenous PP2CA levels were increased in mutants impaired in their respective E3 ligases. Using 35S:PP2CA-HA lines and a CHX chase assay in the presence of ABA, Baek et al. [51] showed that PP2CA levels were increased in the PIR1PIR2 double mutant compared with the WT.

To identify proteins that coimmunoprecipitate with FLAG-PP2CA in Arabidopsis, Julian et al. [26] performed coIP coupled to LC-MS/MS analysis and identified some BTB/POZ and MATH domain proteins (BPMs) as interactors of PP2CA. Interestingly, BPMs are nuclear proteins that function as substrate adaptors of the multimeric CRL3 E3 ligases. In particular, CRL3^BPM^ complexes play a fundamental role in the stress response and hormone signaling through the regulation of different nuclear targets [26,52,53]. PP2CA accumulates predominantly in the nucleus, although interaction of PP2CA with ion transporters of the PM has also been reported [18,54]. BPMs are resident nuclear proteins, whereas RGLG1 translocates from the PM to the nucleus in response to ABA; therefore, different pools of PP2CA might be targeted by these E3 ligases depending on the ABA concentration. In addition to the recruitment of PP2CA, BPM3 and BPM5 also recognize other PP2Cs such as ABI1, ABI2, and HAB1. Moreover, using specific antibodies against ABI1 and HAB1, Julian et al. [26] reported increased levels of HAB1 and ABI1 in the BPM3 BPM5 double mutant compared to WT.

Recently, the previously described LOSS OF GDU2 (LOG2)/ABA-INSENSITIVE RING PROTEIN3 (AIRP3) E3 ligase has also been found to be involved in the regulation of ABI1 levels [55,56,57]. This RING-type E3 ligase is myristoylated, and its membrane localization is important for the interaction with the amino acid exporter regulatory subunit GLUTAMINE DUMPER1 (GDU1) [57]. However, the results of Pratelli et al. [57] do not support that ubiquitination of GDU1 by LOG2 leads to GDU1 degradation; instead, their data suggest that LOG2 activates GDU1 via ubiquitination, and subsequently, the activated GDU1 facilitates amino acid efflux. On the other hand, the AIRP3/LOG2 transcript is upregulated by drought and ABA, and the AIRP3 mutant is impaired in the abiotic stress response [55]. A reasonable explanation for this phenotype could be the lack of ABI1 degradation, as AIRP3 (in concert with UBC27) was recently reported to interact with ABI1 and promote its degradation [56]. The subcellular localization of the AIRP3–ABI1 interaction was not investigated [56]; therefore, it is intriguing to unveil whether AIRP3 follows a similar mechanism to myristoylated RGLG1 and might interact with ABI1 in the PM (as the LOG2-GDU1 interaction) or in the nucleus upon translocation in response to ABA.

## 6. Conclusions

Once the major components involved in the turnover of ABA receptors and PP2Cs have been identified, translational research is feasible to improve drought resistance in crops, for example, by extending the half-life of ABA receptors. Thus, reduced ubiquitylation of ABA receptors might be achieved in (i) knock-out mutants of single or combined E3 ligases, or (ii) ubiquitylation-defective receptors, which could be mutated in certain Lys residues that do not compromise receptor function. Such approaches might lead to longer half-life of ABA receptors and therefore enhanced ABA signaling. For example, detailed structural data obtained for Arabidopsis and crop ABA receptors [58] will allow the modification of certain Lys residues that do not impair ABA binding and inhibition of PP2C activity. These mutations are amenable to genetic engineering in crops by CRISPR/Cas9 genome editing. Additionally, enhanced degradation of PP2Cs can lead to boosting of ABA signaling and enhanced drought tolerance. This can be achieved through stress-inducible overexpression of E3 ligases (or their substrate adaptors) that target PP2Cs. Since the degradation of PP2Cs works in concert with the PYR/PYL/RCAR-mediated inhibition of PP2C activity, a combined approach would have a synergistic effect.

## Figures and Tables

**Figure 1 ijms-22-07103-f001:**
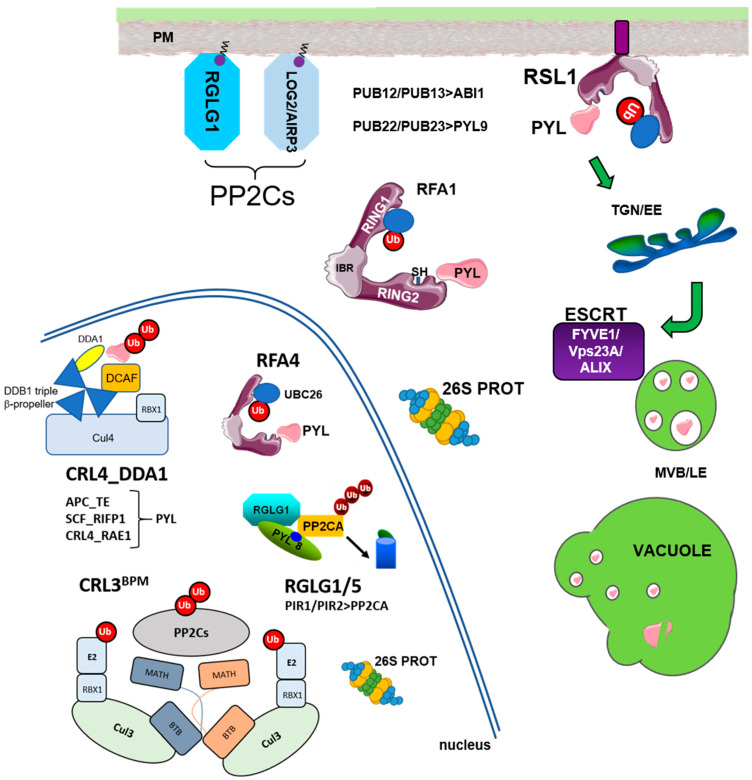
Overview of subcellular localization for E3 ligases that target ABA receptors or clade A PP2Cs showing as well the ESCRT components involved in transit of ABA receptors for vacuolar degradation. Targeting of ABA receptors by RSL1, RFA1, or RFA4 occurs at different subcellular locations. Ubiquitination of ABA receptors on the plasma membrane (PM) by RSL1 triggers clathrin-mediated endocytosis, transit through the early endosomes (TGN/EE), sorting of ubiquitinated cargo through the ESCRT machinery (FYVE1, Vps23A and ALIX), and delivery to multivesicular bodies/late endosomes (MVB/LE) and finally to vacuole for degradation. On the other hand, ubiquitination of receptors and clade A PP2Cs in the cytosol or nucleus leads to degradation by the 26S proteasome. Myristoylated RGLG1 and LOG2/AIRP3 are indicated. SCF, CRL3, CRL4, and APC multimeric E3 ligases are indicated with their corresponding targets as well as monomeric PUB12, 13, 22, 23, and RING-type (RGLG1, AIRP3, PIR1) E3 ligases. Adapted from [11,25,26].

**Table 1 ijms-22-07103-t001:** List of E3 ligases and substrate adaptors of multimeric enzymes that target ABA receptors and clade A PP2Cs. When known, the subcellular localization of the E3 ligase is indicated (PM, plasma membrane). In two cases, the cognate E2 is included in parenthesis.

E3 Ligases_Substrate Adaptor	Localization	Targets	Reference
RSL1	PM	PYR1/PYL4	Bueso et al., 2014
CRL4_DDA1	nucleus	PYL8/PYL9	Irigoyen et al., 2014
APC_TE (rice)	nucleus	OsRCAR10	Lin et al., 2015
SCF_RIFP1	nucleus	PYL8/RCAR3	Li et al., 2016
PUB22/23	cytosol	PYL9/RCAR1	Zhao et al., 2017
CRL4_RAE1	nucleus	PYL9/RCAR1	Li et al., 2018
RFA1/RFA4 (UBC26)	cytosol, nucleus	PYR1/PYL4	Fernandez et al., 2020
PUB12/13	PM for FLS2	ABI1, FLS2, BRI1	Kong et al., 2015 Lu et al., 2011; Zhou et al., 2018
RGLG1/RGLG5	myristoylated PM and nucleus	PP2CA, ABI2, HAB2	Wu et al., 2016 Belda-Palazon et al., 2019
CRL3_BPM	nucleus	PP2CA, ABI1, HAB1, ABI2	Julian et al., 2019
PIR1/PIR2	nucleus	PP2CA	Baek et al., 2019
AIRP3 (UBC27)	myristoylated	ABI1	Pan et al., 2020

**Table 2 ijms-22-07103-t002:** Regulation of ABA receptors’ turnover through the endocytosis/vacuolar degradation pathway. Some ESCRT components that mediate this process are indicated as well as the receptors that were analyzed.

ESCRT Component	ESCRT	Targets Investigated	Reference
FYVE1/FREE1	ESCRT-I	PYR1/PYL4	Belda-Palazon et al., 2016
VPS23A	ESCRT-I	PYR1/PYL4	Yu et al., 2016
ALIX	ESCRT-III	PYL4	Garcia-Leon et al., 2019

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
