# Peer review of "Ubiquitylation of ABA Receptors and Protein Phosphatase 2C Coreceptors to Modulate ABA Signaling and Stress Response"

_ijms, 2021, doi:10.3390/ijms22137103_

Round 1

Reviewer 1 Report

In this review, the authors have comprehensively summarized the ubiquitylation of ABA signaling core components. The review is well written and incorporates the old as well as recent studies done on this topic. I would recommend this review for publication.

Minor comment:

Please write the full name of a given protein/gene at the first instance it is mentioned in the review, before using abbreviations. For example, DDA1 is first mentioned in line 66, while the complete name is given in line 100.

Author Response

Thanks, we have introduced DET1 and DDB1-ASSOCIATED1 (DDA1) in line 66

We have introduced in the text the description of certain abbreviations that are less frequent for readers or more complex to understand. However, to facilitate reading of the text, we do not intend to include exhaustive description of common abbreviations 

Reviewer 2 Report

Dear authors,

I hope you are good.

I read your review about ABA receptors and coreceptors and I think that it is a great contribution to the field of ABA molecular responses. I have not comments about contents. I only have some minor changes:

1.- Write 'post-translational' or 'posttranslational' (see lines 10 and 20). Please, review it through the text.

2.- Table 1. Please, define 'pm'.

3.- I have doubts about the term 'circa'. Could it be substituted by 'approximately' or similar?

Congratulations for your review!

Best regards

Author Response

Thanks a lot!

1- We have written post-translational along the text

2- pm, plasma membrane. Included in Table 1

3- approximately now, we have replaced circa, less common (latin!) 
